# From Voxel to Gene: A Scoping Review on MRI Radiogenomics’ Artificial Intelligence Predictions in Adult Gliomas and Glioblastomas—The Promise of Virtual Biopsy?

**DOI:** 10.3390/biomedicines12092156

**Published:** 2024-09-23

**Authors:** Xavier Maximin Le Guillou Horn, François Lecellier, Clement Giraud, Mathieu Naudin, Pierre Fayolle, Céline Thomarat, Christine Fernandez-Maloigne, Rémy Guillevin

**Affiliations:** 1Laboratoire de Mathématique Appliquées LMA, Labcom i3M, Université de Poitiers, CNRS UMR 7348, F-86000 Poitiers, Francepierre.fayolle@univ-poitiers.fr (P.F.); 2Service de Génétique Médicale, CHU de Poitiers, F-86000 Poitiers, France; 3Laboratoire XLIM, Université de Poitiers, CNRS UMR 7252, F-86000 Poitiers, France; 4Plateforme Ultra-Haut Champ 3T-7T, Laboratoire de Mathématique Appliquées LMA, Labcom i3M, CHU de Poitiers, Université de Poitiers, CNRS UMR 7348, F-86000 Poitiers, Franceremy.guillevin@chu-poitiers.fr (R.G.); 5Laboratoire XLIM, Labcom i3M, Université de Poitiers, CNRS UMR 7252, F-86000 Poitiers, France; christine.fernandez@univ-poitiers.fr; 6Service de Radiologie, CHU de Poitiers, F-86000 Poitiers, France

**Keywords:** adult gliomas, adult glioblastomas, MRI, deep learning, radiogenomics, virtual biopsy, scoping review

## Abstract

Background: Gliomas, including the most severe form known as glioblastomas, are primary brain tumors arising from glial cells, with significant impact on adults, particularly men aged 45 to 70. Recent advancements in the WHO (World Health Organization) classification now correlate genetic markers with glioma phenotypes, enhancing diagnostic precision and therapeutic strategies. Aims and Methods: This scoping review aims to evaluate the current state of deep learning (DL) applications in the genetic characterization of adult gliomas, addressing the potential of these technologies for a reliable virtual biopsy. Results: We reviewed 17 studies, analyzing the evolution of DL algorithms from fully convolutional networks to more advanced architectures (ResNet and DenseNet). The methods involved various validation techniques, including k-fold cross-validation and external dataset validation. Conclusions: Our findings highlight significant variability in reported performance, largely due to small, homogeneous datasets and inconsistent validation methods. Despite promising results, particularly in predicting individual genetic traits, the lack of robust external validation limits the generalizability of these models. Future efforts should focus on developing larger, more diverse datasets and integrating multidisciplinary collaboration to enhance model reliability. This review underscores the potential of DL in advancing glioma characterization, paving the way for more precise, non-invasive diagnostic tools. The development of a robust algorithm capable of predicting the somatic genetics of gliomas or glioblastomas could accelerate the diagnostic process and inform therapeutic decisions more quickly, while maintaining the same level of accuracy as the traditional diagnostic pathway, which involves invasive tumor biopsies.

## 1. Introduction

The brain is primarily composed of neurons, which are responsible for processing information, and glial cells, which provide structural and functional support. Glial cells account for at least 50% to 90% of the brain’s composition [1]. Among them, astrocytes play a critical role in supplying nutrients to neurons, managing interneuronal connections, and regulating processes such as memory, movement, and odor processing [2].

Gliomas are tumors that develop from these glial cells in the brain or spinal cord. Most brain tumors originate in the cerebral hemispheres, particularly in the white matter, which contains a dense network of axons (projections from neurons) surrounded by glial cells. However, gliomas can occur throughout the central nervous system. They are the most common type of brain tumor in children, adolescents, and adults, and are the most prevalent form of primary (i.e., non-metastatic) brain tumor in adults. The most severe form of glioma is glioblastoma, which is also the most common central nervous system tumor in the U.S. population, accounting for 14.2% of all tumors and 50.1% of all malignant tumors. In 2022, there were 13,272 new cases of glioblastoma in the United States, primarily affecting adults and occurring more frequently in men than in women [3].

Over time, the World Health Organization (WHO) classifications have increasingly incorporated correlations between a tumor’s genetic profile and its phenotype. This effort culminated in the 2021 classification, which clearly distinguishes between childhood and adult glial tumors [4]. Adult diffuse gliomas are now categorized into three classes: *IDH1*- and *IDH2*-mutated oligodendrogliomas, which exhibit a 1p/19q codeletion (loss of the p arm of chromosome 1 and the q arm of chromosome 19) and are associated with longer survival, representing low-grade gliomas; *IDH1*- and *IDH2*-mutated diffuse astrocytomas, which have a variable prognosis, ranging from low-grade to high-grade gliomas; and glioblastomas, which lack *IDH1* or *IDH2* mutations but exhibit an activating mutation in the *TERT* promoter (a telomere-building protein), *EGFR* amplification, and specific karyotypic features. These glioblastomas are classified as high-grade gliomas and are associated with the worst prognosis. Additionally, methylation of the *MGMT* gene, which encodes the DNA repair enzyme O-6-methylguanine-DNA methyltransferase, is an important factor in determining the efficacy of temozolomide chemotherapy. If the *MGMT* gene is unmethylated, the cancer cells are capable of repairing the DNA damage induced by temozolomide, reducing the drug’s effectiveness [5,6].

Magnetic resonance imaging (MRI) is considered the gold standard for detecting brain tumors, whether discovered incidentally or as a result of neurological symptoms. MRI can assess the extent of the disease and determine the feasibility of surgical resection. It also guides stereotactic biopsy, the only method capable of providing histological and molecular analyses to confirm the diagnosis and characterize the tumor. This histo-molecular characterization is critical for optimal, patient-specific therapeutic management. However, biopsy is not feasible for all patients, and there is a risk that the sampled area may not represent the most severe or prognostically relevant region of the tumor. Additionally, biopsy is an invasive procedure and carries inherent risks for the patient. Since 2012, the classification of non-medical images has been revolutionized by deep learning (DL) algorithms, particularly those utilizing convolutional neural networks (CNNs) [7]. This technology offers significant potential for application in the genetic characterization of adult glial tumors, presenting a compelling opportunity for non-invasive diagnostics. The development of a robust algorithm capable of predicting the somatic genetics of gliomas or glioblastomas from MRI could accelerate the diagnostic process and inform therapeutic decisions more quickly, while maintaining the same level of accuracy as the traditional diagnostic pathway, which involves invasive tumor biopsies. Furthermore, such an algorithm could provide valuable genetic insights for patients’ ineligible for biopsy, offering a non-invasive alternative for diagnosis and treatment planning.

The aim of our scoping review is to provide a comprehensive, state-of-the-art overview of the use of DL algorithms in the genetic characterization of adult glial tumors. This review will cover reported performance, potential limitations, datasets used (including those available via open access), and future directions needed to advance the field toward achieving a reliable and reproducible capacity for virtual biopsy.

## 2. Materials and Methods

This scoping review was conducted with reference to the Preferred Reporting Items for Systematic Reviews and Meta-Analyses (PRISMA) guidelines, commonly used for systematic reviews [8].

### 2.1. PICOs (Inclusion Criteria)

We defined the PICOs as follows. Population: adults with gliomas or glioblastomas. Intervention: the development and use of AI models to predict somatic genetics from MRI. Comparison: not applicable. Outcomes: the ability to predict genetic features, evaluated using ROC curves. Our focus was on artificial intelligence models utilizing deep learning (DL) technology, rather than traditional machine learning (ML) approaches.

### 2.2. Search Strategy

For our research strategy, we selected three databases: PubMed, Embase, and the Cochrane Library. These databases were chosen for their focus on medical articles relevant to our topic and the management of patients with gliomas. Although Google Scholar was considered, it was not used due to the challenges associated with obtaining complete bibliographic data through scraping methods. IEEE Xplore was also not included, despite its potential relevance due to its coverage of computer science articles that may involve the development of deep learning algorithms for genetic feature prediction. Notably, IEEE Xplore references articles from the IEEE (the Institute of Electrical and Electronics Engineers) and its partners that are often indexed in PubMed.

We established the following criteria for accessing articles: if an article could not be accessed through OpenAccess or Shibboleth library access provided by our institution, it was excluded. Preprints and articles written in languages other than English were also excluded. For PubMed searches, we excluded all references that did not have full-text availability. During the review writing process, we planned to re-query one database for additional relevant articles, choosing PubMed for this second query for simplicity. (query of each databases in Appendix A).

Any article known to us but not included in the search results was included in our review at our discretion. Similarly, if an article cited by one of the query results met the PICOs criteria but was not included in the initial search results, it was also included in our review.

### 2.3. Data Extraction

Search results were compiled using Zotero, an open-access bibliographic tool, which automatically excluded duplicates and irrelevant records. For data extraction, we used a tabular file format (“.csv”).

## 3. Results

The flowchart of the review process is shown in Figure 1. The initial search was conducted across three databases on 30 December 2023, and an additional search on PubMed was performed on 1 June 2024, during the writing of the review. The results of the 17 articles selected for this review are summarized in Table 1. Among these, three articles employed algorithmic methodologies using DL strategies for the segmentation and extraction of radiomic features, which were then processed by ML algorithms such as Support Vector Machine (SVM), k-Nearest Neighbors (k-NN), or Random Forest: S. Rathore et al. [9], S. Kihira et al. [10], and S. Qureshi et al. [11].

### 3.1. Bibliographical and Descriptive Data on Publications

The 17 publications span from 2009 to 2023 (Figure 2a). For each publication, we collected the Scimago Journal Rank (SJR) score for the year of publication, with a median of 1.312, a minimum of 0.297, and a maximum of 4.112. For comparison, the SJR scores in 2023 for major journals were as follows: Nature—18.509, Cell—24.342, and The New England Journal of Medicine—20.544. Among medical imaging journals, Medical Image Analysis had the highest SJR in 2023, with a score of 4.112.

The number of patients used to develop the algorithms ranged from 21 to 985 (Figure 2b). However, I. Hrapșa et al. [21] presented replication work involving 21 patients without retraining, based on the work of Y.S. Choi et al. [20]. In contrast, B.-H. Kim et al. [23] used up to 985 patients, which were combined from the Seoul National University Hospital (SNUH) dataset (400 patients) and the MICCAI BrATS 2021 dataset (585 patients). The algorithm described in their study was initially trained on the 400 patients from the SNUH dataset.

Among the seventeen articles, seven used data from patients with glioblastomas, five used data from patients with gliomas, and five used data from patients with both glioblastomas and gliomas.

The MRI sequences employed in these studies were primarily T2-weighted (T2w) and T2 Fluid-Attenuated Inversion Recovery (T2-FLAIR) (Table 2).

### 3.2. Deep-Learning Strategy

All the articles utilized algorithms that fit the definition of deep learning. Among these, three employed DL algorithms, but the task of classifying tumors based on their genetic characteristics was handled by a separate machine learning (ML) algorithm [2,3,4].

Of the 17 papers, 10 used CNNs with fully convolutional architectures. These CNNs were applied either directly to the images, after automatic segmentation by a deep learning algorithm with a specific architecture, or following manual segmentation.

The specific architectures mentioned included DenseNet, ResNet, and U-Net. One of the studies explored transfer learning by adapting non-medical image classification models for use in medical image classification (H. Sakly et al. 2023 [24]).

### 3.3. Tumor Genetics’ Explored

The articles primarily focused on predicting genetic or epigenetic abnormalities, including the presence of *IDH1/2* mutations, methylation of the *MGMT* promoter, over-expression of *EGFR*, and the 1p/19q co-deletion (Table 3). Additionally, one study investigated the prediction of tumor variations affecting *PTEN*, *ATRX*, *TERT*, *CDKN2A/B*, *TP53*, and chromosomal rearrangements such as Trisomy 7 and Monosomy 10 [9].

#### 3.3.1. IDH Mutation Prediction

Based on the Area Under the Receiver Operating Characteristic (ROC) Curve (AUC), the algorithm proposed by Y.S. Choi et al. [20] achieved the best performance with an AUC of 0.96 (95% CI: 0.93–0.99) (Table 4). However, when this algorithm was replicated on external data by I. Hrapșa et al. [21], the AUC was 0.74 (95% CI: 0.53–0.91), with a sensitivity of 78% and a specificity of 75%. The first study included patients with both gliomas and glioblastomas, while the second focused solely on glioblastoma patients. One of the highest performances was reported by P. Eichinger et al. [13], who achieved an AUC of 0.952 using 79 patients, compared to the 856 patients in the study by Y.S. Choi et al. [20].

#### 3.3.2. MGMT Promoter Methylation Prediction

The highest performance in predicting *MGMT* promoter methylation was achieved by the algorithm of S.A. Qureshi et al. [11], with an AUC of 0.96 (95% CI: 0.94–0.98) (Table 5). This performance was achieved using a deep learning approach for segmentation and radiomics extraction, followed by classification with a Random Forest algorithm, which is a machine learning strategy.

The second-best performance was achieved by P. Chang et al., with an AUC of 0.81 (95% CI: 0.76–0.84) [14]. Two studies reported AUCs with confidence intervals around 0.5, suggesting that their performance may be no better than chance [19,23]. I. Levner’s 2009 study [12],which utilized a small neural network based on MRI texture analysis, achieved an accuracy of 87.7% for predicting *MGMT* promoter methylation. This study, while innovative, is primarily a proof of concept due to its very small dataset. H. Sakly et al. [24] explored transfer learning with image classification algorithms.

Additionally, the work by Saeed et al. [25] did not present the highest performance (Table 5), but was notable for comparing several deep learning algorithm architectures. Their study focused on understanding discrepancies in results from similar algorithms, rather than solely developing the most robust model.

#### 3.3.3. EGFR Amplification Prediction

In our review, only three articles attempted to predict *EGFR* amplification (overexpression) (Table 6). Among these, M. Hedyehzadeh et al. did not present results that could be directly compared, but their deep learning strategy was able to statistically detect *EGFR* amplification in glioblastomas [16].

Of the remaining two studies, S. Rathore et al. achieved slightly better results, although their final classifier was a Random Forest algorithm rather than a deep learning approach [9]. Similarly, E. Calabrese et al. reported better performance with their Random Forest classifier compared to their CNN classifier [22].

#### 3.3.4. Chromosome 1p19q Co-Deletion Prediction

The algorithm with the best performance for predicting 1p/19q co-deletion status was proposed by P. Chang et al. [14] (Table 7). Among the four articles addressing this prediction, B. Kocak et al. [18] tested a CNN and various ML algorithms (k-NN, Random Forest, SVM, etc.) on the same dataset, with the CNN yielding better results than the other methods.

## 4. Discussion

### 4.1. Evolution of Publications

A chronological analysis of the publications reveals a significant gap between the initial study in 2009 and the subsequent surge in research starting from 2017. Several factors may account for this hiatus. In 2009, glioma and glioblastoma classification was primarily based on histological criteria, with limited integration of genetic characteristics into therapeutic decision-making. At that time, the methylation status of the *MGMT* promoter was the primary genetic factor influencing chemotherapy choices.

I. Levner et al. [12] conducted a proof-of-concept study in 2009 using a two-layer CNN to predict MGMT promoter methylation with 87% accuracy on a training dataset. It was not until 2017 that more sophisticated research emerged, incorporating advanced CNN architectures such as ResNet and DenseNet. This shift reflects advancements in DL technologies and the increasing importance of genetic factors in glioma classification and treatment.

### 4.2. Deep Learning Algorithms

The neural network architectures employed in the studies reviewed have evolved significantly over time. Early studies utilized Fully Convolutional Neural Networks (FCNNs). Later, more advanced architectures such as ResNet, DenseNet, and U-Net were introduced. Some researchers also explored transfer learning using models like AlexNet, GoogleNet, and InceptionV2 [24].

The use of DL algorithms for the genetic characterization of glial tumors from medical imaging is well justified by the extensive application of these algorithms in computer vision tasks. Deep learning has become a cornerstone in the classification and analysis of common objects, providing highly detailed and nuanced representations through distributed parameterization. These DL algorithms can capture complex patterns in imaging data that traditional methods might miss. While some studies indicate that traditional ML methods may outperform DL strategies in predicting specific genetic characteristics (S. Rathore et al. [9] or S. A. Qureshi et al. [11]), the key question is not whether ML is superior to DL, but rather under which conditions and architectural configurations DL algorithms will fully surpass ML approaches for this task. The focus should be on identifying the optimal DL architectures and settings that can achieve superior performance in genetic characterization, rather than comparing DL and ML as broad categories.

However, there was considerable variation in the validation methods used across studies. While some studies employed a classic train–test split, others used k-fold cross-validation with varying k-values (3, 4, or 5) or Leave-One-Out Cross-Validation (LOOCV) for small sample sizes. External validation on independent datasets was notably infrequent, which could be a significant limitation.

Notably, Y. S. Choi et al. conducted an external validation of both the TCIA and SNUH datasets [20]. Their study revealed significantly poorer results on the TCIA dataset compared to the SNUH dataset, which showed comparable results. This disparity may be attributed to differences in patient management and imaging protocols between Seoul and the institutions contributing to the TCIA dataset. The consistency in management and imaging modalities, as well as histological and molecular studies, within the SNUH dataset could explain the better performance on this dataset.

### 4.3. Performance and Reproducibility

Algorithms with very high AUCs (close to 1) raise questions about their generalizability. A publication by N. Saeed et al. [25] explored this issue, highlighting that the limited size of datasets and inadequate validation methods can artificially inflate performance. Most studies are based on a single dataset, thus limiting the reproducibility of results.

The reproducibility of these algorithms, based on this literature review, is largely questionable. Many studies report exceptional performance metrics, however, these results often derive from specific datasets with limited external validation. The lack of robust testing across diverse datasets and clinical settings raises concerns about the reliability of these algorithms in real-world healthcare applications. For these deep-learning models to be applicable in clinical practice, it is imperative to enhance their robustness and generalizability. Future research must focus on improving the validation processes and expanding datasets to ensure that these algorithms can consistently perform well across various populations and imaging conditions. Only through such advancements can we hope to make these algorithms reliably exploitable in healthcare settings.

Although some attempts have been made to predict multiple genetic traits within a single model, results are often presented individually for each trait. This suggests a juxtaposition of classifiers rather than an integrated model capable of distributed prediction in a multidimensional space.

### 4.4. Limitations and Challenges

The main limitations identified include the limited size and homogeneity of datasets. With fewer than 1000 patients, these datasets may not capture the full variability needed for robust model training, particularly given that a single DICOM MRI scan can yield up to 1000 radiomics parameters. Additionally, the frequent lack of well-defined external validation datasets undermines the assessment of algorithm reliability and reproducibility, which are crucial for ensuring model generalizability.

An important area for improvement is the integration of an explanatory dimension into algorithm development. This would facilitate a better understanding of algorithm limitations and biases, which is crucial for ensuring medical liability. Without such transparency, the deployment of these algorithms in clinical settings could face significant legal and regulatory challenges, potentially hindering their future use in healthcare.

Furthermore, addressing the genetic heterogeneity of tumors presents a significant challenge. The current datasets are inadequate for this purpose; therefore, there is a need to develop datasets incorporating three-dimensional information from biopsy areas, potentially in the form of a three-dimensional biopsy probability field.

Another challenge is the variability between datasets, as exemplified by differences in results between the TCIA and SNUH datasets reported by Y. S. Choi et al. [20]. The lack of standardized guidelines for MRI assessment of brain tumors—such as sequence types and field strength (3 T versus 1.5 T)—and the variability in genetic data acquisition and analysis due to factors like neurosurgical expertise, histological study, and molecular biology methods further complicate the situation.

To advance this field, it is crucial to foster interdisciplinary collaboration among mathematicians, radiologists, pathologists, geneticists, and neurosurgeons to effectively address clinical demands and overcome these challenges.

## 5. Conclusions

Our review of 17 articles on the application of deep learning (DL) algorithms in the radiogenomic characterization of gliomas and glioblastomas underscores both the advancements and the persistent challenges in this emerging field. Since the early studies in 2009, and particularly since 2017, significant progress has been made. However, numerous obstacles remain to ensure the effectiveness and generalization of these models for a potential virtual biopsy.

The increased interest in this field since 2017 is attributed to technological advancements and a growing recognition of the importance of genetic markers in the classification, prognosis, and treatment of gliomas and glioblastomas, as reflected in the WHO 2021 classification. However, the methodologies for model validation and the limited size of datasets continue to be major concerns. The seemingly high performance of some algorithms must be approached with caution due to the frequent lack of external validation, raising questions about their generalizability.

Efforts to integrate the prediction of multiple genetic traits into a single model are still in the early stages. Most studies focus on predicting individual traits, indicating a reliance on separate classifiers rather than a comprehensive, multidimensional approach. To address this, it is essential to develop larger and more diverse datasets, incorporating clinical history, ethnic diversity, tumor characteristics, and molecular and cytogenetic features. ITo account for tumor heterogeneity, and incorporating three-dimensional information on the areas of probability of biopsy could further enhance our understanding of genetic variation within tumors and improve the assessment of tissue characteristics at tumor resection margins.

The limitations of current datasets and validation methods underscore the need for interdisciplinary collaboration. Integrating expertise from mathematicians, radiologists, pathologists, geneticists, and neurosurgeons is crucial for developing models that are both reliable and clinically relevant. Additionally, incorporating an explanatory dimension into algorithms is vital for understanding and addressing their biases and limitations, which is essential for overcoming potential legal and regulatory hurdles.

### Key Recommendations for Future Research

External Validation: Ensure robust validation on external datasets to assess generalizability and avoid overfitting on internal datasets, which may artificially inflate performance.Improved Validation Methods: Apply advanced validation techniques such as k-fold cross-validation with sufficiently large k-values and Leave-One-Out Cross-Validation (LOOCV), especially for small datasets, to improve reliability.Dataset Size and Diversity: Use larger and more diverse datasets, capturing clinical, genetic, and demographic variability (e.g., tumor types, patient populations, and ethnicities) to ensure broad applicability of the algorithms.Integrated Multi-Genetic Trait Models: Focus on developing integrated models capable of predicting multiple genetic traits simultaneously, rather than separate classifiers, to better reflect the complexity of gliomas.Integration of Tumor Heterogeneity: Develop models that take account of tumor heterogeneity to improve the understanding of tumor complexity.Explanatory and Interpretable AI: Ensure that deep learning models include interpretable components to allow clinicians to understand algorithm predictions, thus enhancing their trust in AI tools and ensuring accountability in clinical settings.Standardized MRI Acquisition Protocols: Establish standardized protocols for MRI acquisition (e.g., field strength and sequence types) to reduce variability between datasets and improve model reproducibility.Cross-Disciplinary Collaboration: Promote interdisciplinary collaboration among data scientists, radiologists, pathologists, geneticists, and neurosurgeons to design clinically relevant models that align with real-world clinical workflows.Ethical and Legal Frameworks: Address ethical and regulatory considerations, ensuring that the developed models comply with standards for medical liability, data privacy, and patient safety, especially given the potential future deployment of AI in healthcare.

Finally, although DL holds promise for advancing the radiogenomic characterization of gliomas and glioblastomas, ongoing improvements in datasets, validation methods, and interdisciplinary co-operation are necessary to fully realize the clinical potential of these tools. Such advancements will be instrumental in achieving objective early therapeutic decision-making and aligning with the principles of personalized, preventive, predictive, participative, and pertinent (5P) medicine.

## Figures and Tables

**Figure 1 biomedicines-12-02156-f001:**
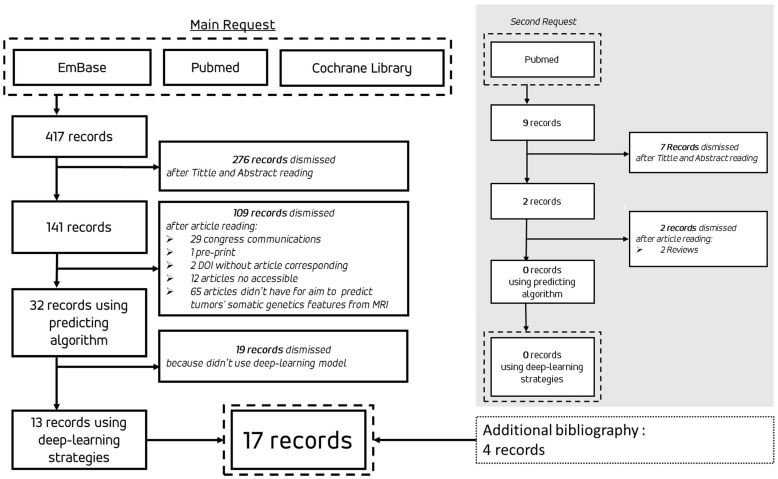
Flowchart of the review process from initial search to selected records.

**Figure 2 biomedicines-12-02156-f002:**
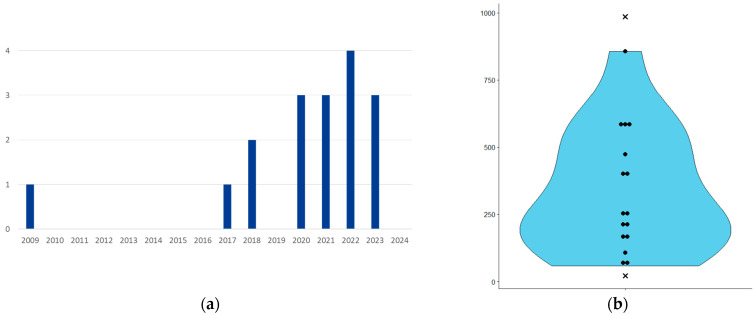
(**a**) Publications per year; (**b**) violin plot showing the number of patients used in algorithm training and testing. The cross points correspond to 21 patients (I. Hrapșa et al. [21]) and 985 patients (B.-H. Kim et al. [23]). *x*-axis: no unit, *y*-axis: number of patient.

**Table 1 biomedicines-12-02156-t001:** Summary of the 17 articles sorted by year of publication. SJR: Scimago Journal Ranking; TCIA: The Cancer Imaging Archive; TCGA: The Cancer Genome Atlas; BrATS: Brain Automated Tumors Segmentation; SNUH set: Seoul National University Hospital dataset.

Authors	Year	Journal	SJR Pub. Year	Title	Number of Patients	Gliomas/ Glioblastomas/ Both	MRI Modalities	Dataset	Algorithms
I. Levner et al. [12]	2009	Medical Image Computing and Computer-Assisted Intervention	0.297	Predicting MGMT Methylation Status of Glioblastomas from MRI Texture	59	Glioblastomas	T1-Gd, T2, T2-FLAIR	Local	CNN (2 layers)
P. Eichinger et al. [13]	2017	Scientific Reports	1.533	Diffusion tensor image features predict *IDH* genotype in newly diagnosed WHO grade II/III gliomas	79	Gliomas	T2-FLAIR	TCIA	N-net
P. Chang et al. [14]	2018	AJNR Am J Neuroradiol	1.543	Deep-Learning Convolutional Neural Networks Accurately Classify Genetic Mutations in Gliomas	259	Gliomas	T1w, T1-Gd, T2w, T2-FLAIR	TCIA, TCGA	CNN
S. Liang et al. [15]	2018	Genes	1.592	Multimodal 3D DenseNet for *IDH* Genotype Prediction in Gliomas	167	Both	T1w, T1-Gd, T2w, T2-FLAIR	BrATS-2017, TCGA	M3D-DenseNet
M. Hedyehzadeh et al. [16]	2020	Journal of Digital Imaging	1.055	A Comparison of the Efficiency of Using a Deep CNN Approach with Other Common Regression Methods for the Prediction of EGFR Expression in Glioblastoma Patients	166	Glioblastomas	T1w, T1-Gd, T2w, T2-FLAIR	TCIA, TCGA	CNN
Y. Matsui et al. [17]	2020	Journal of Neuro-Oncology	1.256	Prediction of lower-grade glioma molecular subtypes using deep learning	217	Gliomas	T1w, T2w, T2-FLAIR, Spectrometry, PET scan	Local	ResNet into CNN
B Kocak et al. [18]	2020	European Radiology	1.606	Radiogenomics of lower-grade gliomas: Machine Learning-based MRI texture analysis for predicting 1p/19q codeletion status	107	Gliomas	T1w, T2w	TCIA	CNN against ML algorithms
S Rathore et al. [9] *	2020	Neuro-Oncology Advances	1.052	Multi-institutional non-invasive in vivo characterization of *IDH*, 1p/19q, and EGFRvIII in glioma using neuro-Cancer Imaging Phenomics Toolkit (neuro-CaPTk)	473	Both	T1w, T1-Gd, T2w, T2-FLAIR, DSC, DCE	Local, TCIA, TCGA	Neuro-CaPTK (Cancer Imaging Phenomics Toolkit)
C. G. B. Yogananda et al. [19]	2021	AJNR Am J Neuroradiol	1.34	MRI-Based Deep-Learning Method for Determining Glioma MGMT Promoter Methylation Status	247	Gliomas	T2w	TCIA, TCGA	3D-dense-Unets
Y. S. Choi et al. [20]	2021	Neuro-Oncology	3.097	Fully automated hybrid approach to predict the *IDH* mutation status of gliomas via deep learning and radiomics	856	Both	T1w, T2w, T2-FLAIR	Local, SNUH set, TCIA	CNN
I. Hrapșa et al. [21]	2022	Medicina	0.59	External Validation of a Convolutional Neural Network for *IDH* Mutation Prediction	21	Glioblastomas	T1w, T2w, T2-FLAIR	Local, TCIA, TCGA	CHOI et al.’s CNN [20]
E. Calabrese et al. [22]	2022	Neuro-Oncology Advances	1.052	Combining radiomics and deep convolutional neural network features from preoperative MRI for predicting clinically relevant genetic biomarkers in glioblastoma	400	Glioblastomas	T1w, T2w, T2-FLAIR, SWI, DWI, ASL, MD, AD, RD	Local	CNN Limb
B.-H. Kim et al. [23]	2022	Cancers	1.312	Validation of MRI-Based Models to Predict MGMT Promoter Methylation in Gliomas: BraTS 2021 Radiogenomics Challenge	400 (+585)	Both	T1w, T1-Gd, T2w, T2-FLAIR	Local, SNUH set, BrATS 2021	Efficient-Net, squeeze-and-excitation networks, SEResNet, SEResNeXt, DenseNet
S. Kihira et al. [10] *	2022	Cancers	1.312	U-Net Based Segmentation and Characterization of Gliomas	208	Both	T2-FLAIR	Local	DenseNet121
H. Sakly et al. [24]	2023	Cancer Control: Journal of the Moffitt Cancer Center	0.698	Brain Tumor Radiogenomic Classification of O6-Methylguanine-DNA Methyltransferase Promoter Methylation in Malignant Gliomas-Based Transfer Learning	585	Glioblastomas	T1w, T1-Gd, T2w, T2-FLAIR	BrATS 2021	Alexnet, Googlenet, Resnet, ImageNet, VGG, DenseNet, Xception, InceptionV3Squeezenet
S. A. Qureshi et al. [11] *	2023	Scientific Reports	0.9	Radiogenomic classification for MGMT promoter methylation status using multi-omics-fused feature space for least invasive diagnosis through mpMRI scans	585	Glioblastomas	T1w, T1-Gd, T2w	BrATS 2021	CNN for segmentation and extraction feature but SVM or k-NN for classification
N. Saeed et al. [25]	2023	Medical Image Analysis	4.112	MGMT promoter methylation status prediction using MRI scans. An extensive experimental evaluation of deep learning models	585	Glioblastomas	T1w, T1-Gd, T2w, T2-FLAIR	BrATS 2021	ResNet, DenseNet, EfficientNEt

* Articles employing deep learning strategies did not advance to the genetic characterization of tumors.

**Table 2 biomedicines-12-02156-t002:** MRI sequences used in articles employing DL algorithms. T1w: T1-weighted; T1-Gd: T1-weighted post-gadolinium contrast; T2w: T2-weighted; T2-FLAIR: T2 Fluid-Attenuated Inversion Recovery.

MRI Sequence	Number	Percent
T1w	13	76%
T1-Gd	9	53%
T2w	14	82%
T2-FLAIR	14	82%
Spectrometry	1	6%
Other	3	18%

**Table 3 biomedicines-12-02156-t003:** Genetic and epigenetic abnormalities explored in the studies.

Genetic Features	Number	Percent
*IDH1*/*2* mutation	9	53%
*MGMT* methylation	9	53%
*EGFR* expression	3	18%
1p19q codeletion	4	24%
Other	1	6%

**Table 4 biomedicines-12-02156-t004:** Prediction of *IDH* mutation presence. Includes AUC (Area Under the ROC Curve); 95% confidence intervals (CI) are shown in brackets where available. NA: Not Available.

	Tumors Type	AUC (95% CI)	Accuracy (%)	Sensitivity (%)	Specificity (%)
P. Eichinger et al. [13]	Gliomas	0.952	0.95	NA	NA
P. Chang et al. [14]	Gliomas	0.91 (0.89–0.92)	NA	NA	NA
S. Liang et al. [15]	Both	0.857	84.6	78.5	88.0
Y. Matsui et al. [17]	Gliomas	NA	82.9	NA	NA
S. Rathore et al. [9]	Both	0.87	82.5	70.43	88.32
Y.S. Choi et al. [20]	Both	0.96 (0.93–0.99)	93.8	NA	NA
I. Hrapșa et al. [21]	Glioblastomas	0.74 (0.53–0.91)	76	78	75
E. Calabrese et al. [22]	Glioblastomas	0.96 (0.88–1)	84	100	83
S. Kihira et al. [10]	Both	0.93 (0.90–0.97)	NA	0.98	0.32

**Table 5 biomedicines-12-02156-t005:** Prediction of *MGMT* promoter methylation. AUC (Area Under the ROC Curve), 95% confidence intervals (CI) are shown in brackets where available. NA: Not Available.

	Tumor Type	AUC (95% CI)	Accuracy (%)	Sensitivity (%)
I. Levner et al. [12]	Glioblastomas	NA	87.7	85.4
P. Chang et al. [14]	Gliomas	0.81 (0.76–0.84)	NA	NA
C. G. B. Yogananda et al. [19]	Gliomas	0.58 (0.4182–0.7422) ^1^	65.95	NA
E. Calabrese et al. [22]	Glioblastomas	0.73 (0.65–0.81) ^2^	68	72
B.-H. Kim et al. [23]	Both	0.517 (0.459–0.645)	51.9	NA
S. Kihira et al. [10]	Both	0.62 (0.54–0.71)	NA	0.45
H. Sakly et al. [24] ^3^	Glioblastomas	NA	NA	NA
S. A. Qureshi et al. [11]	Glioblastomas	0.96 (0.94–0.98) ^4^	96.94	96.31
N. Saeed et al. [25]	Glioblastomas	0.631 (0.629–0.633)	NA	NA

^1^ Results of C. G. B. Yogananda et al. include erratum: https://doi.org/10.3174/ajnr.A7715. ^2^ Combined method using CCN and Random Forest algorithm on radiomics feature achieved an AUC of 0.77 (95% CI: 0.63–0.91). ^3^ Did not present results on one or more validation datasets. ^4^ Achieved best performance but utilized a final step of ML classifier.

**Table 6 biomedicines-12-02156-t006:** Prediction for *EGFR* amplification. AUC (Area under the ROC curve) and 95% confidence interval (CI) are shown in brackets where available. NA: Not Available.

	Tumor Type	AUC (95% CI)	Accuracy (%)	Sensitivity (%)	Specificity (%)
M. Hedyehzadeh et al. [16] ^1^	Glioblastomas	NA	NA	NA	NA
S. Rathore et al. [9]	Both	0.80 ^2^	86.74	84.91	87.5
E. Calabrese et al. [22]	Glioblastomas	0.72 (0.64–0.80) ^3^	66	68	66

^1^ No metrics could be used as comparison. ^2^ Trained on 248 patients, with the maximum of 473 patients available in the study. ^3^ Appeared less performant than the Random Forest classifier on radiomics (AUC = 0.77 [95% CI: 0.67–0.87]) and combined classifier (AUC = 0.80 [95% CI: 0.74–0.86]).

**Table 7 biomedicines-12-02156-t007:** Prediction for co-deletion of chromosomes 1p and 19q. AUC (Area Under the ROC Curve) and 95% confidence interval (CI) are shown in brackets where available. NA: Not Available.

	Tumor Type	AUC (95% CI)	Accuracy (%)	Sensitivity (%)	Specificity (%)
P. Chang et al. [14]	Gliomas	0.88 (0.85–0.90)	NA	NA	NA
Y. Matsui et al. [17]	Gliomas ^1^	NA	75.1	NA	NA
B. Kocak et al. [18]	Gliomas	0.869 (0.751–0.987) ^2^	83.8	87.5	75.8
S. Rathore et al. [9]	Both	0.79 ^3^	75.15	81.49	73.96

^1^ Presented patients with *IDH* wild-type gliomas, which have been classified as glioblastomas since 2021. ^2^ Both the CNN and ML algorithms trained on the same dataset. ^3^ Trained on 192 patients, with a maximum of 473 patients available in the study.

## Data Availability

No new data were created or analyzed in this study. Data sharing is not applicable to this article.

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
