# Peer review of "From Voxel to Gene: A Scoping Review on MRI Radiogenomics’ Artificial Intelligence Predictions in Adult Gliomas and Glioblastomas—The Promise of Virtual Biopsy?"

_biomedicines, 2024, doi:10.3390/biomedicines12092156_

Round 1

Reviewer 1 Report

Comments and Suggestions for Authors

This paper “From Voxel to Gene: a Scoping Reviews on MRI Radiogenomics Artificial Intelligence Predictions in Adult Gliomas and Glioblastomas - the Promise of Virtual Biopsy?”

The topic is justified. The paper could be further improved if the following remarks are taken into consideration:

1.       ABSTRACT: implications of the research is missing in the abstract section.

2.       Full stop be after reference 3 in second paragraph of the introduction section.

3.       A few of the grammatical mistakes were found in the whole draft of the article; the authors need to fix these.

4.       Introduction section: key contributions need to be defined.

5.       Research questions need to be defined explicitly.

6.       Figure 1 is blurry.

7.       The answers to the defined research questions be established under th light of reviewed literature.

8.       The motivation is not clear.

9.       Discussion section needs more analysis.

10.   Revise the conclusions. Go beyond the summary of works done. Enlist the specific advantages over similar methods. What are the implications of these studies for further research in the domain?

Comments on the Quality of English Language

minor edits required

Author Response

Comment 1: ABSTRACT: implications of the research is missing in the abstract section.

Response: Motivations have been added to the abstract

Comment 2: Full stop be after reference 3 in second paragraph of the introduction section.

Response: Corrected

Comment 3: A few of the grammatical mistakes were found in the whole draft of the article; the authors need to fix these.

Response: Draft corrected by a native American

Comment 4: Introduction section: key contributions need to be defined.
Response: We didn't understand whether we should expand on the references concerning the general context of computer vision.

Comment 5: Research questions need to be defined explicitly.
Response: The motivations for this literature review and its objectives are detailed in the last 2 paragraphs of the introduction.

Comment 6: Figure 1 is blurry.

Response: Not in our version of draft (resolution 1766*1099 pixels)

Comment 7: The answers to the defined research questions be established under th light of reviewed literature.
Response: They seem to meet our objectives set out in the introduction. So as not to make the discussion too long, we'll focus on those elements of the articles which we feel are most important for defining “Key Reccomendations” (for example, we won't discuss the datasets accessible in detail). However, we have expanded our analysis in the discussion section.

Comment 8: The motivation is not clear. 

Response: In the introduction and conclusion, we reiterated the motivation for the review, namely “what are the challenges involved in enabling a virtual biopsy of a glial tumor using a deep learning algorithm based on MRI as input data?

Comment 9: Discussion section needs more analysis.

Response: Cf response of comment 8

Comment 10: Revise the conclusions. Go beyond the summary of works done. Enlist the specific advantages over similar methods. What are the implications of these studies for further research in the domain?

Response We have added 9 Key reccomendation for future research in this specific field

Reviewer 2 Report

Comments and Suggestions for Authors

Xavier LE GUILLOU HORN et al manuscript clearly mention the use of deep learning in the genetic characterization of adult gliomas.

Minor comments:

1) In the introduction part, the author needs to first introduce gliomas and their clinical significance, followed by genetic characterization.

2) Ensure the language is concise with clarify

3) The author needs to mention why only PubMed, Embase and Cochrane library was selected, why not other databases like IEEE Xplore.

Comments on the Quality of English Language

english correction needs to be done.

Author Response

Comment 1: In the introduction part, the author needs to first introduce gliomas and their clinical significance, followed by genetic characterization.

Response: That's waht we do

Comment 2: Ensure the language is concise with clarify

Response: the draft was corrected by American native whose  work was to limited emphasis.

Comment 3:The author needs to mention why only PubMed, Embase and Cochrane library was selected, why not other databases like IEEE Xplore.

Response: We have added a justification, particularly on IEEExplore 
